# Flights of a Multirotor UAS with Structural Faults: Failures on Composite Propeller(s)

**Srikanth Gururajan** [1,*], **Kyle Mitchell** [2] and **William Ebel** [2]

[1] Aerospace Engineering, Parks College of Engineering, Aviation and Technology, Saint Louis University, St. Louis, MO 63103, USA

[2] Electrical and Computer Engineering, Parks College of Engineering, Aviation and Technology, Saint Louis University, St. Louis, MO 63103, USA

* Correspondence: srikanth.gururajan@slu.edu; Tel.: +1-314-977-8355

**Abstract:** Data acquired from several flights of a custom-fabricated Hexacopter Unmanned Aerial System (UAS) with composite structure (carbon fiber arms and central hub) and composite (carbon fiber) propellers are described in this article. The Hexacopter was assembled from a commercially available kit (Tarot 690) and flown in manual and autonomous modes. Takeoffs and landings were under manual control and the bulk of the flight tests was conducted with the Hexacopter in a "position hold" mode. All flights were flown within the UAS flight cage at Parks College of Engineering, Aviation and Technology at Saint Louis University for approximately 5 min each. Several failure conditions (different types, artificially induced) on the composite (carbon fiber) propellers were tested, including failures on up to two propellers. The dataset described in this article contains flight data from the onboard flight controller (Pixhawk) as well as three accelerometers, each with three axes, mounted on the arms of the Hexacopter UAS. The data are included as supplemental material.

**Keywords:** hexacopter; UAS structural failures; UAS flight tests; failures in composite materials; UAS fault tolerance; Pixhawk

---

## 1. Introduction

Drones or Unmanned Aerial Systems (UAS) are ubiquitous in today's world. They are used as toys, platforms for commerce, or as vehicles for the testing and validation of advanced research topics in various scientific fields. There have been numerous instances of drones used in civilian applications such as aerial photography for various applications [1,2], crop monitoring [3,4], infrastructure assessment [5], as well as in disaster recovery [6], law enforcement, and many other applications.

These operations are predicated on the assumption that they are safe and that the UAS platform is reliable and can be controlled by the operator at all times, including autonomous operations. Given the fact that these UAS are mechanical systems and they are subject to wear and tear from operational use, it is inevitable that faults or failures will occur. Therefore, in order to ensure that the flights of drones in all these domains remain safe for the operator and airspace, it is critical to study their behavior under failure or fault conditions and to facilitate the better development of tools to mitigate the effects of such failures. The origins of failures in drones can be numerous, ranging from manufacturing errors to in-flight failures. Under both conditions, it is important to address slowly evolving failures, as they do not manifest into readily observable behavior until, in most cases, the structural failures result in

unrecoverable conditions. In recent years, studies have been conducted on the use of embedded [7,8] or surface-mounted sensing elements on the structure of the UAS to observe and characterize structural failures. In another work [9], failures on the aircraft structure, including the propeller, were studied by utilizing external accelerometers mounted on the arm(s) of the quadcopter, in a static configuration.

At the Aircraft Computation and Resource Aware Fault Tolerance (AirCRAFT) Lab [10] at Saint Louis University, St. Louis, MO, we designed experiments to address one aspect of such failure conditions on the drone—a broken propeller. This data descriptor article describes the setup of the UAS, the sensors onboard, the failure conditions, and the flight test experiments.

These experiments were conducted as a preliminary, unfunded study to establish the baseline performance of the UAS under failure conditions. Our intent is to build upon this set of flight tests to generate an increasingly rich set of failures and under various flight conditions as well as to design, develop, and validate flight control algorithms that can detect, identify, and accommodate for such failures, so as to ensure safe drone flights in the National Airspace System (NAS). Beyond our own research, we believe that this dataset could benefit the broader scientific research community in several ways, including but not limited to those listed below:

i.    Facilitate collaborative efforts in the study of the long-term evolution of structural failures on a UAS.
ii.   Provide a foundation for multidisciplinary efforts on the development of mitigation approaches to such failures, especially when these drone platforms are being/will be used for commercial applications such as package delivery.

## 2. Data Description

This article describes the data collected from controlled tests of a custom built Hexacopter UAS, conducted at the AirCRAFT Laboratory. The following subsections describe the airframe, the flight controller/data acquisition, the individual sensors, the flight data (including details on the data), its format, and how to read the data.

*Hexacopter Unmanned Aerial System*

The UAS platform used in this experiment is a Hexacopter, shown in Figure 1. It was built from a widely available kit from Tarot-RC [11] and its specifications are listed in Table 1. The schematic in Figure 2 shows the position of the accelerometers mounted on alternate arms of the hexacopter. Figures 3 and 4 illustrate the geometrical distances between various components on the UAS.

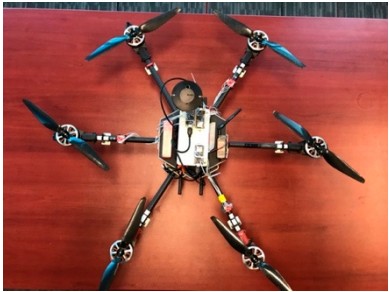

**Figure 1.** Tarot-RC 690 Hexacopter Unmanned Aerial System (UAS).

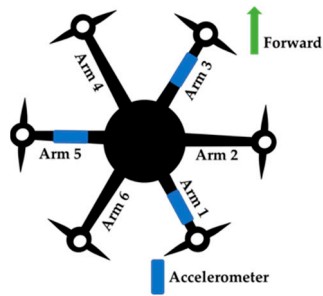

**Figure 2.** Schematic of Hexacopter UAS with accelerometers mounted on the arms.

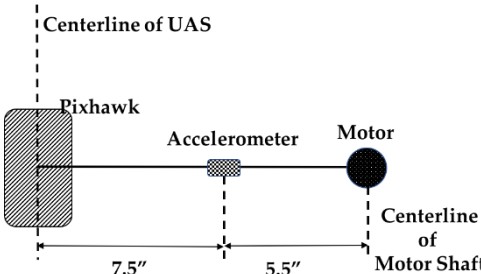

**Figure 3.** Location of motor, accelerometer, and Pixhawk; top view of UAS.

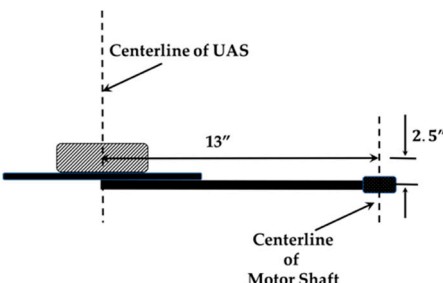

**Figure 4.** Location of motor, accelerometer, and Pixhawk; side view of UAS.

**Table 1.** Specifications of the Tarot-RC 690 Hexacopter UAS.

| Parameter | Specification |
| --- | --- |
| Airframe | Tarot-RC 690 Hexacopter |
| Motors and ESC | TYI-Power 5008, 335 Kv; Lumenier 40A ESC |
| Propellers | Composite (Carbon fiber 12″ × 5.5″) |
| Flight Controller/Firmware | Pixhawk 2.1 [12]; |
| All Up Weight | 8.1 lb (~3.68 Kg) |
| Battery/Endurance | 6 s, 5800 mAh LiPo battery/ approx. 8 min |
| Transmitter/Receiver | Taranis X9D/Hitech Minima 6 Channel |
| Telemetry | Holybro 915 MHz |

Onboard Flight Controller, Flight Mode and Data Acquisition

The Hexacopter is integrated with a wide available Pixhawk 2.1 flight cotroller and a GPS unit. The Pixhawk flight controller supports several flight modes, including a "position hold" mode. This mode [13] enables the UAS to maintain its location, heading, and altitude. In this mode, and under relatively constant environmental conditions, the effort of the flight controller is relatively constant and the artifacts seen in the acceleration data could be attributed to the changes in the physical conditions of the UAS—nominal and failure. A sample of the flight data, including the 3D GPS track of the UAS

in position hold, is shown below in Figure 5, along with the output of the controller (inputs to the ESC of each motor) in Figure 6.

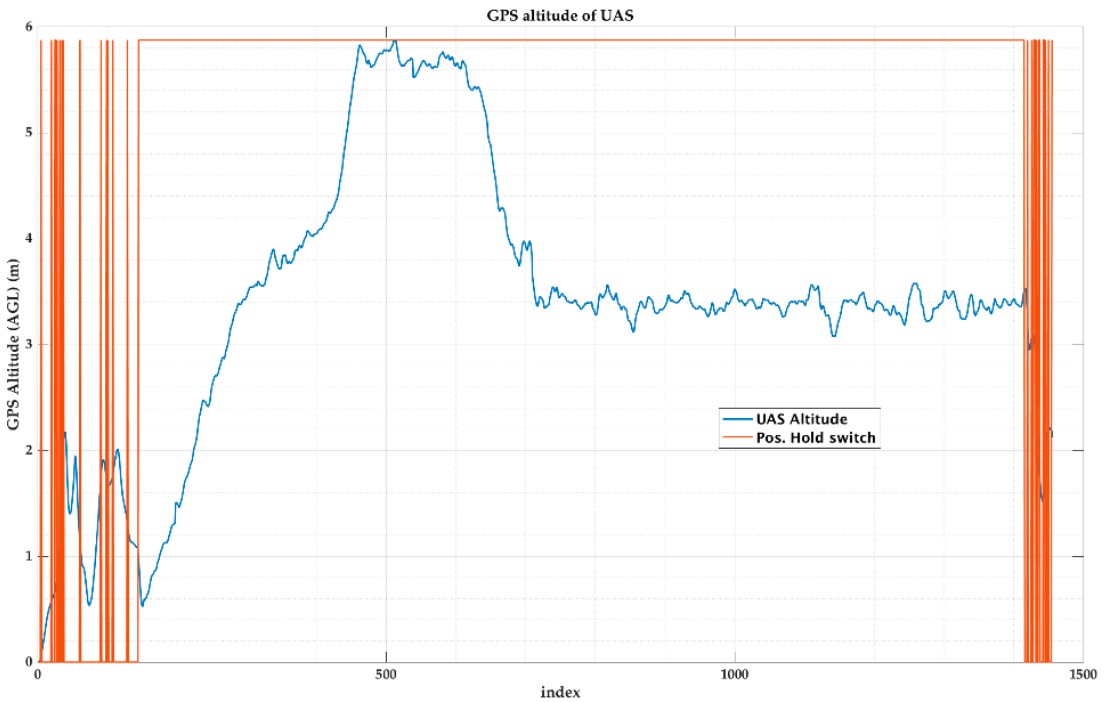

**Figure 5.** GPS altitude above ground level of Hexacopter UAS.

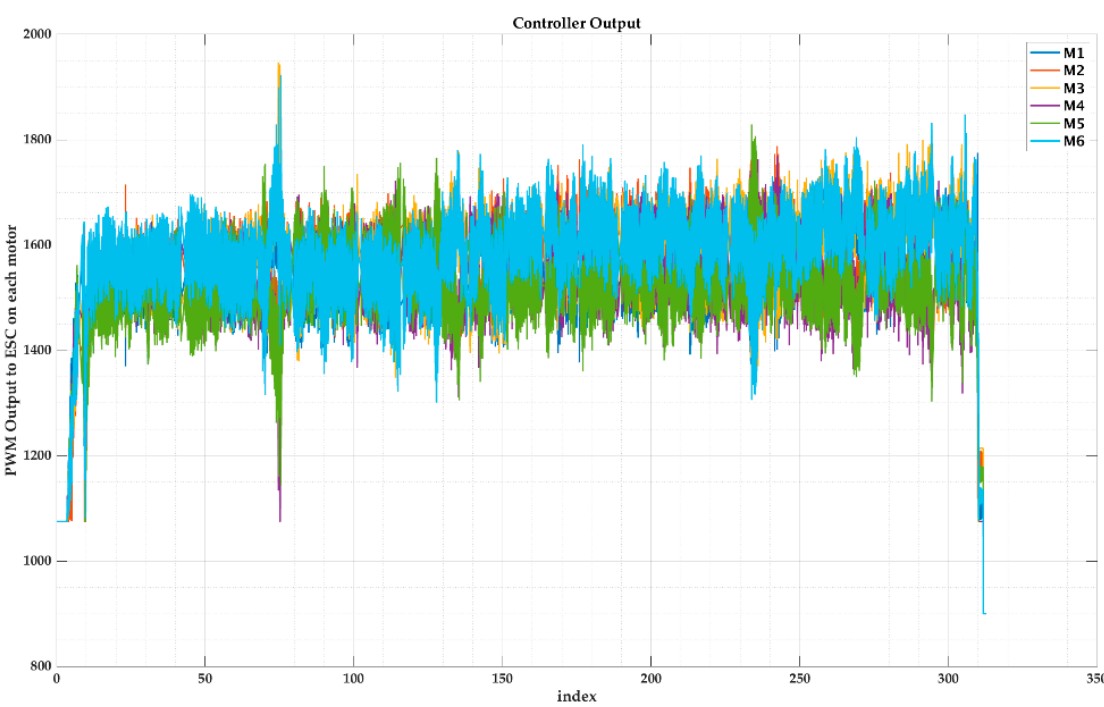

**Figure 6.** Output of the flight controller, raw Pulse Width Modulation (PWM) values.

## 3. Methods

### 3.1. Flight Tests

The Hexacopter was flown in both manual and autonomous "position hold" mode in the outdoor UAS flight cage at Parks College of Engineering, Aviation and Technology at Saint Louis University. The flight cage (shown in Figure 7) has dimensions of approximately $150 \times 40 \times 40'$. This test facility is designed with soccer netting on all sides to accommodate flights of small multi-rotor drones in the real-world environment, but within a safe enclosure.

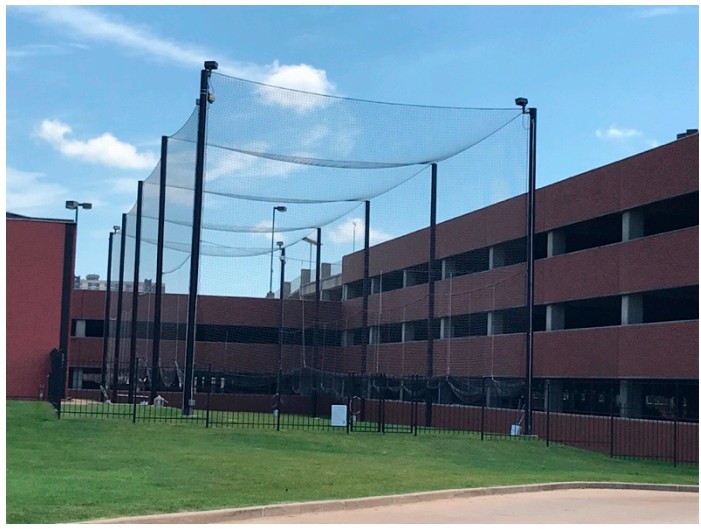

**Figure 7.** Outdoor UAS flight cage at Parks College of Engineering, Aviation, and Technology, Saint Louis University.

The datasets described in this article were collected from the flights of the Hexacopter UAS within this cage. The UAS took off in manual mode and, once it reached a certain altitude, it was switched to an autonomous "position hold" mode that was programmed in the flight controller. The total duration of each flight of the UAS, including the "position hold" mode, was approximately 5 min before the landing sequence was initiated—a switch back in the flight mode from "position hold" to "stabilize" and manual landing. The data from the onboard flight controller is also attached to this article for use by the wider scientific community. Flights were flown in the following four different scenarios described in Table 2 below.

**Table 2.** Flight test scenarios.

| | |
|---|---|
| *i.* | Nominal flight |
| *ii.* | Failure conditions on propeller |
| | *a.* Failure condition 1: Failure on one propeller (sensor arm) |
| | *b.* Failure condition 2: Failure on one propeller (arm adjacent to sensor arm) |
| | *c.* Failure condition 3: Failure on two propellers (sensor arm and adjacent arm) |

The failure conditions refer to damages to the propellers on motors #3 and #4 of the Hexacopter. For the purposes of this experiment, the propellers were willfully damaged. The damaged propellers are shown in Figures 8 and 9, respectively.

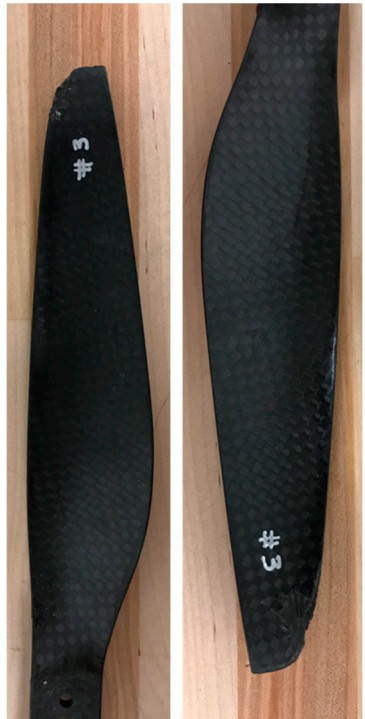

**Figure 8.** Broken propeller #3.

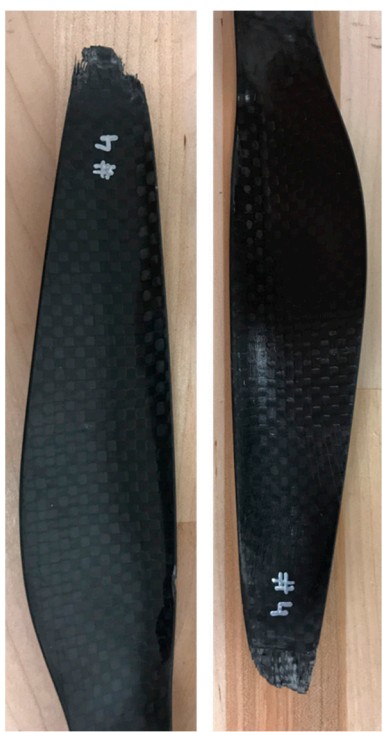

**Figure 9.** Broken propeller #4.

*3.2. Data Acquisition from External Accelerometers and Processing*

The Hexacopter was outfitted with three ST H3LIS331DL three-axis digital accelerometers [5], as shown in Figure 10. These accelerometers were placed on every other arm (#1, #3, and #5) and orientated such that their *x*-axis pointed along the arm, from the central hub outward to the motor and their *z*-axis pointed upward. The accelerometers were configured to sample at 1 kHz using their

internal oscillator and to have a full range measurement of ±100 g. The sensors have a 16 bit, two's complement output, giving them a quantized precision of $3.05 \times 10^{-3}$ g. The raw data were converted to g by dividing by 327.68. A few salient details of the accelerometers [5] are listed below in Table 3. Additional details can be found in [14].

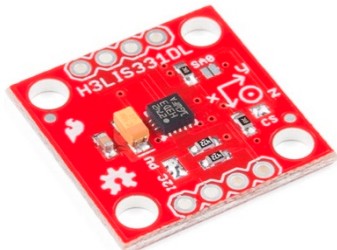

**Figure 10.** ST H3LIS331DL three-axis accelerometer.

**Table 3.** Salient specifications of the arm-mounted accelerometers.

| Parameter | Specification |
| --- | --- |
| Input voltage | 2.16 V–3.6 V |
| Power consumption | 10 µA in low power mode |
| Range | ±100, ±200, ±400 g, dynamically selectable |
| Output | 16 bit data output |
| Output data rates | 5 Hz to 1 kHz |
| Operating temperature | −40 °C to +85 °C |

The ST H3LIS331DL was wired to an Adafruit Feather M0 Adalogger [15], as shown in Figure 11.

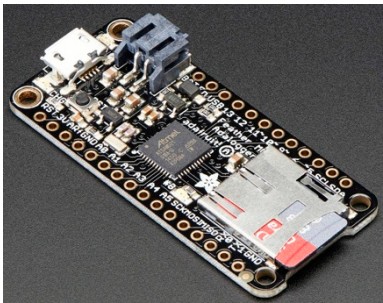

**Figure 11.** Featherboard M0 datalogger.

The Feather M0 [15] is an ATSAMD21G18 ARM Cortex M0-based single-board computer. The Adalogger is an Arduino-programmable device that contains a microSD connector. The Adalogger watches for a conversion complete signal from the accelerometer. It then reads the three axes and writes a data packet to the SD card. The data packet contains the following fields:

- Node number,
- USB frame number,
- The Arduino microsecond timer value,
- A packet index,
- The *X*, *Y*, *Z* tuple,
- A Carriage Return, Line Feed (CRLF)

The USB frame number, Arduino microsecond timer value, and packet index are used to detect missing data values and to synchronize samples amongst the sampling nodes. Since all the nodes

are recording the USB frame number as a common beacon, samples can be synchronized around these values.

A dataset that has a packet index sequence that increases by more than 1 between samples and is assumed to have experienced data loss in the path between the processor and SD card. A dataset that shows a microsecond timer jump of more than 500–1500 ms between samples is assumed to have missed data between the accelerometer and the processor. In what we consider to be a good dataset, all the timer increments should be between 980 and 1020 ms, due to the way the Arduino interrupts work in conjunction with reading the microsecond timer.

### 3.3. Data Resampling and Synchronization

The fact that data are being collected based on three separate oscillators (on the three accelerometers on the arms of the Hexacopter) leads to three different sample rates and the need to resample the data so as to synchronize the three data channels.

The recording of the USB frame number generated by the Raspberry Pi Zero W places a common beacon in the data streams. These beacons cannot be used directly due to the variance between the sample clocks and the USB frame number rate. The microsecond timer values cannot be used to synchronize either as they have different start times and are generated by oscillators the exist separate from each other and the data.

To facilitate synchronization, synthetic time vectors were produced for each data channel by using the USB frame numbers to calculate the elapsed time and the data length to calculate a sample period. This arrangement assumes that the data are sampled with an underlying constant sample period. It also assumes that the USB frame number is regular and equally spaced. Analysis has shown that one of these assumption is incorrect. The synthetic time vector plotted against the frame number sequence is shown in Figure 12.

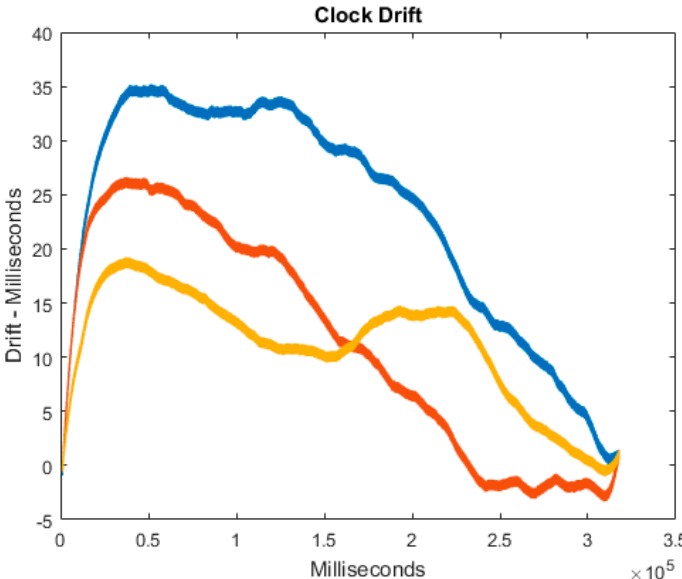

**Figure 12.** Clock drift among the three accelerometers on the Hexacopter.

This chart indicates that either the USB frame numbers are not evenly spaced or the accelerometer samples are not evenly spaced. For this dataset, the accelerometer sample spacing was chosen to be considered as the more accurate of the two.

To synchronize the samples, a resample process is used. A common target time vector is produced that contains a synthetic 1 ms sample period and time values that are common to synthetic times vectors of all three datasets. A convolutional resample process is then used to produce new data points for each of the sample sets at the new 1 kHz sample rate. This convolutional resample process also

includes a low pass filter. The cutoff frequency for this filter was set at 450 Hz. It is important to note that the first 5 s of all the datasets were discarded, as analysis has shown that the PI Zero/Linux modifies the USB frame number at several points in the first 3.5 s.

Figure 13 shows the processed accelerometer data (*x*-direction, along the arm) from a nominal flight (without failure) and a flight with one broken propeller (on motor #3).

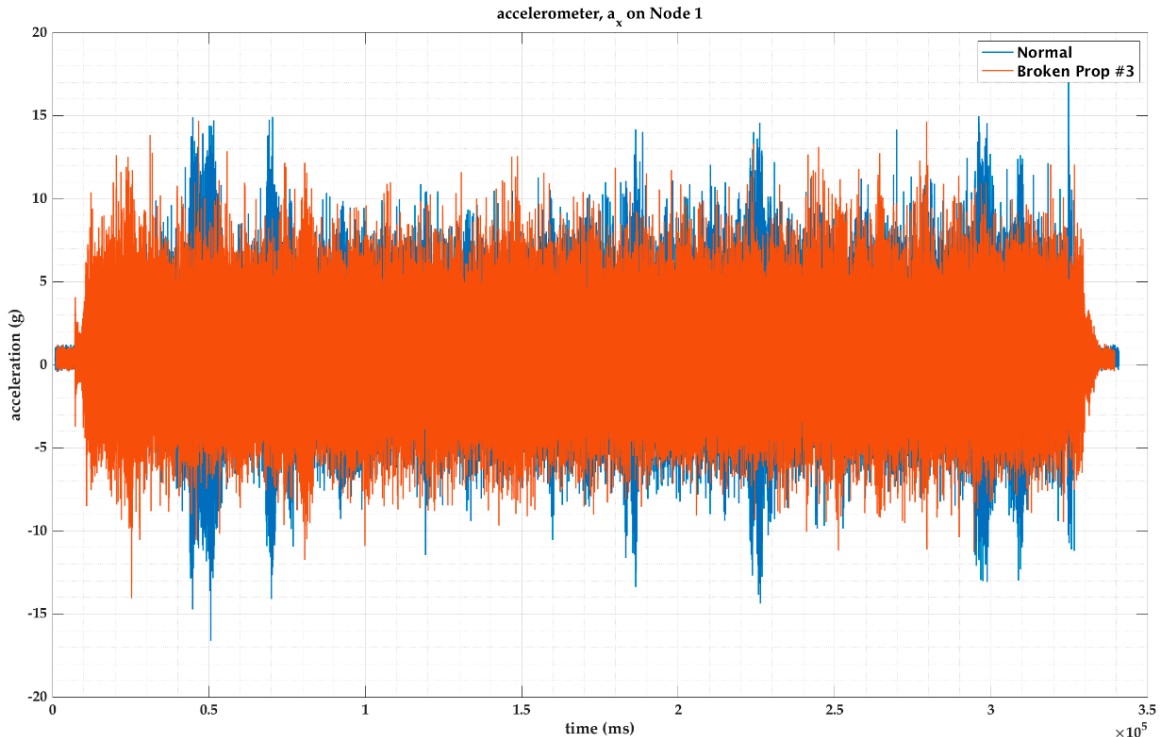

**Figure 13.** Accelerometer output ($a_x$) from one node (n1) under nominal and failure conditions.

## 4. User Notes—Description of the Data Channels

The data are stored in two locations. The accelerometer data are stored in a file named "Accelerometer Data-FC1Flt01", indicating that this contains data from failure condition #1 and its corresponding flight, Flight 01. This is a MATLAB ".mat" file which contains three data structures named n1, n2, and n3, representing data from each instrumented node. Each of these data structures contains the time vector and the data from the three accelerometers $a_x$, $a_y$, and $a_z$. To plot the data, the reader could issue a simple command(s) in MATLAB, as listed in Table 4 below.

**Table 4.** Simple MATLAB commands to plot accelerometer data from the Hexacopter UAS.

| |
|---|
| load "Accelerometer Data-FC1Flt01.mat"; % this loads the data into the workspace figure; plot (n1.Time, n1.ax); % this plots the $a_x$ data, with time as the independent variable hold on plot (n1.Time, n1.ay); % this plots the $a_y$ data, with time as the independent variable plot (n1.Time, n1.az); % this plots the $a_z$ data, with time as the independent variable |

The data from the Pixhawk flight controller are stored as a ".csv" file in its appropriate folder. For the user to plot the data, they would have to import these data into the MATLAB workspace using the "Import Data" function. The files all contain headers for each column of data that are self-explanatory and easy to use.

## 5. Concluding Remarks

This article describes the data collected from controlled flights of a custom fabricated Hexacopter UAS. The UAS was flown under two distinct failure conditions: failure of a single propeller of the Hexacopter and failure of two propellers of the Hexacopter. The data can be used to further analyze the vibration signature of the Hexacopter for the detection, identification, and characterization of slowly evolving failures on dynamic systems—in this particular case, a drone.

**Author Contributions:** Conceptualization, S.G. and K.M.; methodology, S.G. and K.M.; software, K.M. and W.E.; validation, S.G., K.M., and W.E.; formal analysis, K.M.; investigation, S.G. and K.M.; resources, S.G. and K.M.; data curation, S.G. and K.M.; writing—original draft preparation, S.G. and K.M.; writing—review and editing, S.G., K.M., and W.E.; visualization, S.G. and K.M.; supervision, S.G. and K.M.; project administration, S.G. and K.M.

**Funding:** This research received no external funding.

**Acknowledgments:** The authors would like to thank members of the AirCRAFT lab, Jeremy Glowacki, Micah Dubach, Sagar Calnoor, and Matt Dreyer, for all their efforts in support of the experiments.

**Conflicts of Interest:** The authors declare no conflict of interest.

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
