# Peer review of "Flights of a Multirotor UAS with Structural Faults: Failures on Composite Propeller(s)"

_data_

Round 1
Reviewer 1 Report
The paper presents a dataset that is collected from the flight data of an onboard flight controller and three 3-axis accelerometers mounted on the arms of a Hexacopter. The flight data was collected from a well-known open-source flight controller, Pixhawk. The dataset is useful to be used to study the effects of broken propellers on the overall drone behavior.
Major remarks
- The Introduction section is missing, I think it is somehow inside the Summary section.
- The State-of-the-art section needs to be included to give the readers an overview of the related work in this area (e.g., flight controllers, drone failures, etc.).
- The conclusion section needs to be added.
Minor issues:
- Several typos in the paper. For example: “data set and dataset; position hold mode [4] the it enables...”.
- American vs British English. For example Artefacts.
Author Response
The Introduction section is missing, I think it is somehow inside the Summary section.
Author response:
We acknowledge the oversight. The “summary” section was intended to be an introduction. This has been corrected in the revision. We have also extended the Introduction to include relevant literature on applications of drones, and studies on failures on drones.
The State-of-the-art section needs to be included to give the readers an overview of the related work in this area (e.g., flight controllers, drone failures, etc.).
Author response:
We have updated the article, with relevant details about widely used COTS flight controllers, such as Pixhawk (all its variants), and other avionics hardware. We have also included information from relevant literature on failures on drones (particularly structural failures)
The conclusion section needs to be added.
Author response:
We have updated the manuscript to include a section with concluding remarks.
Minor issues:
Several typos in the paper. For example: “data set and dataset; position hold mode [4] the it enables...”.
American vs British English. For example Artefacts.
Author response:
We thank the reviewer for catching the inconsistency. We have made editorial revisions, and are following a consistent way to address the sets of data as “dataset”.
We have corrected the spelling to “artifact”, to be consistent with American spelling.
Reviewer 2 Report
1. No literature review at all, only websites. Please add some drone related literature. 2. Fig 1 photo do not correspondence with fig 2 (draw). Accelerometers are marked on the different arms (different than on the photo) 3. Table 2. Provide more detailed information connected with sensing instruments, especially provide relative position form center of centerplate, where are accelerometers and Pixhawk IMU units (provide drawing). This is very important for data analysis. Provide weight of the drone. 4. Provide detailed technical data of sensors used (in table form). 5. Provide conclusions. 6. Do some proofreading.
Author Response
No literature review at all, only websites. Please add some drone related literature
Author response:
We have updated the manuscript to include relevant references to literature on drones their applications, and efforts on characterization of structural failures on UAS
Fig 1 photo do not correspondence with fig 2 (draw). Accelerometers are marked on the different arms (different than on the photo)
Author response:
We thank you for catching the misalignment. The illustration is now fixed to correspond to the picture, with the accelerometers located on the corresponding arms.
Table 2. Provide more detailed information connected with sensing instruments, especially provide relative position form center of centerplate, where are accelerometers and Pixhawk IMU units (provide drawing). This is very important for data analysis. Provide weight of the drone.
Author Response
We have included two illustrations with the details of the locations of the accelerometers, with respect to the Pixhawk, as well as the motor, as well as an illustration highlighting the vertical distance between the Pixhawk and the accelerometers.
We have updated the UAS specifications table with the all up weight of the UAS.
Provide detailed technical data of sensors used (in table form).
Author response:
We have included the salient specifications of the 3-axis accelerometer in a table format, as requested. We have also included a note pointing the reader to additional, more extensive pdf documentation on the sensor, from the manufacturer.
Provide conclusions.
Author response:
We have updated the manuscript to include a section with concluding remarks.
Do some proofreading.
Author response:
We appreciate all the reviewer’s inputs on editorial aspects of our article. We have made every effort to fix typos, and inconsistencies in language.
Reviewer 3 Report
I am ok with publishing this manuscript.
Author Response
I am ok with publishing this manuscript.
Author response:
We thank the reviewer for taking time to read through our article, and appreciate your comments.